# Mathematical Model of Air Dryer Heat Pump Exchangers

**Tomasz Mołczan** *[ID] **and Piotr Cyklis** [ID]

Faculty of Mechanical Engineering, Cracow University of Technology, al. Jana Pawla II 37, 31-864 Cracow, Poland
* Correspondence: tomasz.molczan@doktorant.pk.edu.pl

**Abstract:** This paper presents a mathematical model of heat pump exchangers and their thermal interaction with a fan for an air dryer. The calculation algorithm developed for the finned heat exchangers is based on the $\varepsilon$-NTU method, allowing the determination of air side and refrigerant side heat transfer coefficients, evaporator and condenser heat capacity and air parameters at the dehumidifier outlet with known exchanger geometries, initial air parameters and mass flow rate. The model was verified on an industrial dehumidifier test bench. This enabled the heat transfer coefficients for the exchanger to be calculated as a function of the speed and, therefore, the power of the fan's drive motor. An increase in fan performance on the one hand results in an increase in the heat transfer rate, but, on the other hand, it causes an increase in the total energy consumption of the motor. Thus, while it causes an increase in drying capacity, it also causes an increase in the energy consumption of the dehumidifier. In order to optimise the unit in terms of energy consumption, it is therefore necessary to determine a function that relates the amount of heat exchanged to the efficiency of the fan.

**Keywords:** air dryer; heat pump; finned tube heat exchanger; condenser; evaporator; fan–exchanger interaction



## 1. Introduction

Drying is one of the most energy-consuming processes used in many industries, including: food production, woodworking, sewage treatment, papermaking and the sanitation of clothes [1–4]. There are several drying methods, but the most common is thermodynamic drying using a heat pump and a refrigerant cycle. An important aspect of reducing the energy consumption in processes such as this is the appropriate selection of components, especially those used in the heat pump and the optimised continuous control of the fan and compressor. For this purpose, it is necessary to develop dependencies between the capacity of the heat exchangers and the efficiency and power demand of the fan based on the simulation model. The most essential elements of the heat pump are heat exchangers—especially for the closed drying cycle [5]. In these exchangers (Figure 1), the dried air is cooled below the dew point, then heated and returned to the drying chamber. The air is cooled by the cold evaporator, and it is reheated by the hot condenser of the cooling cycle [6].

Finned heat exchangers are the most commonly used type of heat exchanger in fan air dryers. Their basic components are stainless steel or copper pipes with elbows, collectors and distributors, aluminium or copper fins supporting the heat exchange process and the housing [5,7]. To intensify heat transfer, it is necessary to use materials with good thermal conductivity and to maximise heat transfer on the walls of heat exchanging elements [8] (pp. 225–227). There are many publications describing the possibilities of intensifying heat transfer on the air side, especially due to the fact that, as it has been shown in research [9], 90% of the total thermal resistance is represented by the values from this (air) side. Among the available articles, it is possible to distinguish two groups of publications: the first investigating the effect of increasing the surface of the fins [10–12] and the second increasing turbulence through the vortex generators on the fins [13–15]. For example, in one publication [12], wavy fins were evaluated, it was estimated that they could improve

heat transfer by up to 70%, but, on the other hand, it caused increases in pressure losses on the air side by up to 140% compared to standard fins.

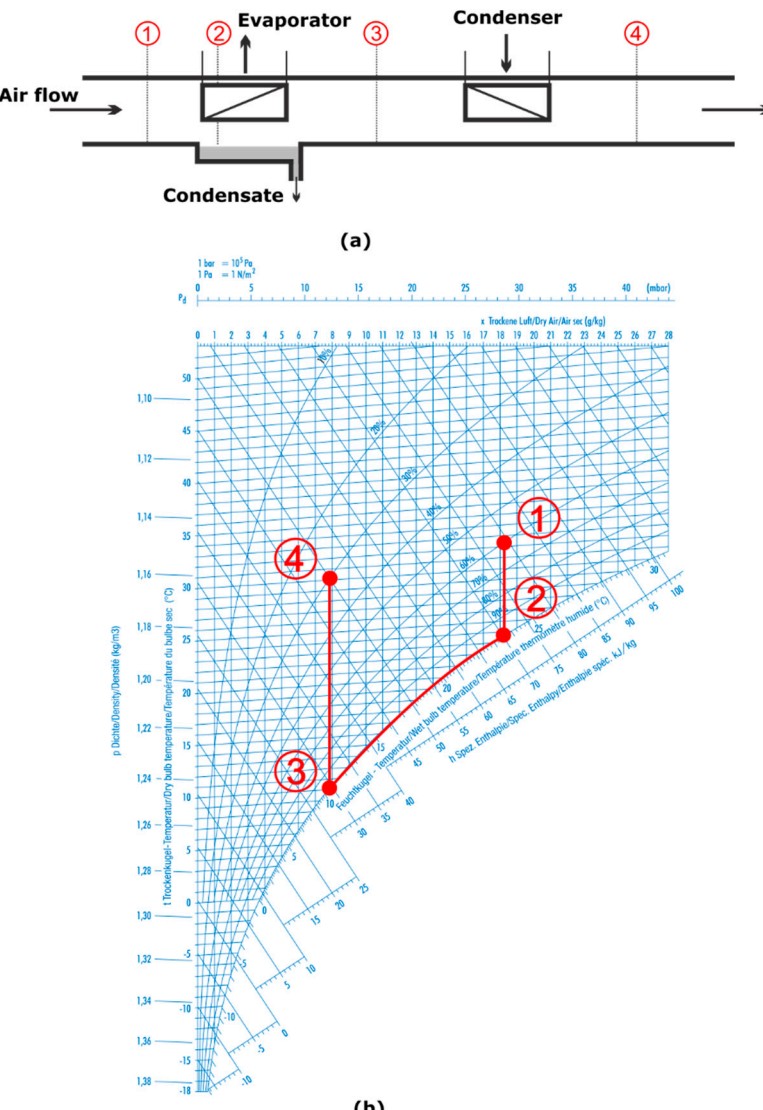

**Figure 1.** Diagram of the drying process (**a**) and the drying process in the psychometric graph (**b**).

Since this type of exchanger is commonly used, there are many publications concerning these cases and mathematical models used for thermal calculations and flow resistance for single-fin heat exchangers [16–21]. The algorithms that enable determination of the thermal efficiency of heat exchangers with known design parameters, calculation examples and criteria dependencies are described in [16]. Sekulić and Shah [20] reviewed the methodology of designing heat exchangers, presenting the basis of design and calculations. The $\varepsilon$-NTU method used in this article can be applied to heat calculations of the exchanger. The relationships and calculation examples concerning the heat and mass transfer in fin exchangers are also presented in one of the chapters of the extensive publications [21].

In one article [17], the authors proposed a model for the evaporation of melted water from the evaporator fins flowing into a container placed on the top of a compressor, which was then positively verified by an experimental test. This can be used as a pre-design tool for the entire condensate evaporation system. In another publication [18], Zalewski presented a method of conducting thermal calculations of a fin heat exchanger, the effect of which is to determine the heat exchange surface area with a known computational efficiency. In another article by this author [19], he presented a calculation algorithm for

the thermal calculations that could be applied, inter alia, in the case of fin air coolers with a fan forcing the flow.

In study [22] on the impact of pipe geometry on the change of the efficiency of the fin exchanger on the air side, an algorithm for heat transfer between humid air and the refrigerant was presented as well as equations to estimate the efficiency of the exchanger using the Chilton–Colburn coefficient. The topic of heat transfer in an exchanger with finned tubes is also described in a book by Hesselgrave, Law and Reay [23]. It includes graphs of the Colburn coefficient and the friction coefficient depending on the Reynolds number for various pipe geometries, as well as an explanation of the NTU method with dependencies on the efficiency of the exchanger. Another article [24] examines the effect of the distance between the exchanger fins on heat exchange, the pressure drop on the air side and the accumulation of frost. A model was developed to compare different fin spacings while maintaining the evaporator geometry.

In other research [11], the authors considered the problem of freezing water on the exchanger. Of the three configurations proposed, the best results were obtained with an evaporator characterised by a gradual reduction in the fin distance on successive rows of tubes. The choice of the right distance at the design stage can make a significant difference in the subsequent correct operation of the unit operating at sub-zero evaporation temperatures. Additionally, protective coatings can be used on the elements of fin heat exchangers, e.g., a hydrophobic coating, the use of which has been analysed in studies [25]. It has been proven that they are beneficial in the wet operating conditions of the evaporator, where there is water vapor condensation on the cold elements. With an additional coating, the water droplets tend to run off, which slows down the process of ice accumulation on the surface of the evaporator.

The effect of circulating air volume on the circulating air temperature, coefficient of performance, moisture extraction rate and specific moisture extraction rate were experimentally investigated in [26]. Among other things, the results showed the effect of air circulation on energy consumption. The energy effects associated with crop drying and the influence of air volume were also pointed out by the authors [27–29]. The authors [30] showed the effect of compressor capacity on drying effects but did not address the problems of other flow machines. In [31] the drying kinetics and performance of the open, closed and partially open heat pump drying systems were compared. In [32] the novel technology of an air cycle heat pump drying system is presented with comparison, simulation and experimental investigation. In [33,34] multistage heat pump systems for drying are presented. The authors [35] analyse drying with and without recirculation, which also regulates the air flow to some extent, although without affecting the fan power. In each case, the authors of the mentioned works pay attention to the energy consumption and the influence of the dried air flow on the process. However, they do not refer to the possibility of controlling the fan and linking it to the operation of the heat exchanger.

As shown above, the calculation methods for heat exchangers are known and well verified. Nevertheless, they are not sufficient to optimise the cooperation of the fan (or fans) with the exchangers and, consequently, with the entire heat pump system. Changing the fan speed affects the heat-exchange process in the exchanger by increasing or reducing the amount of heat transferred by changing the heat transfer coefficient. At the same time, the change in speed causes a change in energy consumption by the drive motor of the fan. Therefore, in order to optimise the process of the drying cycle, it is necessary to experimentally verify the function of the dependence of the heat transfer flux on air velocity based on the exact calculation dependencies for the evaporator and the condenser of the heat pump.

The innovation of the presented method lies in the fact that, based on well-verified calculation formulas for convective heat transfer on the air side, a relationship has been developed that relates fan power and heat exchanger performance under various operating conditions. The presented method allows optimization of the entire unit, i.e., adjustment of

the ventilator power to the actual heat exchanger capacity. In this way, electricity consumption for the drive is reduced while maintaining the drying capacity of the heat pump.

## 2. Thermal Calculation Algorithm

The model described in this paper uses the NTU (number of transfer units) method, the calculations of which require specific exchanger and flow parameters, determined at the design stage. The following assumptions were made to develop the model:

- The exchanger works in a steady state (with constant flow rates and temperatures of air and thermodynamic medium).
- Heat losses to and from the environment are taken into account in the calculations.
- Structural elements do not absorb or generate thermal energy.
- At each point in the exchanger cross-section a uniform temperature distribution is assumed.
- The thermal resistance of each element is constant and uniform throughout its volume.
- The refrigerant in the exchangers only undergoes a phase-to-phase change, and sub-cooling and superheating are not considered.
- Heat transfer coefficients between fluids are independent of temperature, time and location.
- The mass flow of air and refrigerant at the inlet is the same as at the outlet, and the flow rate is evenly distributed through the exchanger throughout the volume.
- The air flow direction corresponds to the orientation of the fins.
- Thermal radiation is not taken into account in the calculations.
- The thermophysical properties of the fluids and the exchangers are constant.
- The resistance (thermal and hydraulic) of the water vapor condensing from the air on the cold elements of the evaporator is disregarded.

The described method is related to the efficiency of the exchanger, which can be defined as the ratio of the actual heat flux to the maximum

$$NTU = \frac{k_{Aw} * A_w}{\dot{W}_p} \tag{1}$$

$$\varepsilon = 1 - e^{-NTU} \tag{2}$$

The described method is related to the efficiency of the exchanger, which can be defined as the ratio of the actual heat flux to the maximum

$$\dot{Q} = \varepsilon * \dot{W}_p * \left(t_{p1} - t_o\right) \tag{3}$$

We use this method either when the inlet and outlet temperatures of the refrigerant are known, or when we can determine them from the energy balance. In this case, the formula for calculating the thermodynamic efficiency can be used as the ratio of the temperature differences

$$\varepsilon = \frac{t_{p1} - t_{p2}}{t_{p1} - t_o} \tag{4}$$

If only the inlet temperatures are known, this method requires an iterative procedure. To determine the efficiency, it is necessary to define the maximum heat flux depending on the type of fluid flow—co-current or countercurrent. Additionally, apart from sensible heat, the latent heat, which occurs when the moisture content from the air condenses on the cold elements of the exchanger, should be considered. In Equations (1) and (3) above, there is the heat capacity of air flux $\dot{W}_p$, which considers the occurrence of latent heat

$$\dot{W}_p = \dot{m}_p * c_{pp} * RCJ \tag{5}$$

The ratio of total heat to sensible heat (RCJ) is equal to 1 when cooling or heating air without drying or humidifying it—without any latent heat transfer.

Additionally, Equation (1) defines the heat transfer coefficient related to the internal heat exchange surface area $A_w$, which can be described by the equation

$$k_{A_w} = \frac{1}{\frac{1}{\alpha_c} * \frac{A_w}{A_w} + \frac{\delta_r}{\lambda_r} * \frac{A_w}{A_m} + R_z + \frac{A_w}{\alpha_p * (A_r + \varepsilon_{\dot{z}} * A_{\dot{z}})}} \qquad (6)$$

The individual indexes of $A$ values in the above formula can be defined as: $m$—pipes related to the average diameter, $r$—external surface area of the pipe between the lamellas, $\dot{z}$—fins. The thermal calculation algorithm can be divided into calculations focusing on the determination of the heat transfer coefficient from the air side and the refrigerant side, cooling capacity and air outlet parameters. The components of Equation (6) are discussed below.

### 2.1. Heat Transfer Coefficient from Air Side $\alpha_p$

The determination of the heat transfer coefficient from the air side refers to any equation relating to the flow around a bundle of fin pipes, which should be adapted to the discussed case

$$\alpha_p = \frac{\lambda_p}{d_z} * Nu_p \qquad (7)$$

The Nusselt number can be determined using the Schmidt formula [21]

$$Nu_p = K * Re_p{}^{0.6} * \left(\frac{A_c}{A_{gl}}\right)^{-0.15} * Pr_p{}^{1/3} \qquad (8)$$

In the above Equation (8), the total surface area of the finned and non-finned tube $A_c$ and the surface area of the smooth tube $A_{gl}$ were considered. The $K$ parameter depends on the number of pipes and the pattern of the exchanger used. The individual values are shown in Table 1 (below).

**Table 1.** K parameter values from Equation (8) [21].

| Type Arrangement and Number of Rows | K Parameter |
| --- | --- |
| In-line arrangement, 1–3 rows | 0.20 |
| In-line arrangement, above 4 rows | 0.22 |
| Staggered arrangement, 2 rows | 0.33 |
| Staggered arrangement, 3 rows | 0.36 |
| Staggered arrangement, above 4 rows | 0.38 |

By inserting Equation (8) into Equation (7), we obtain the expression

$$\alpha_p = \frac{\lambda_p}{d_z} * K * Re_p{}^{0.6} * \left(\frac{A_c}{A_{gl}}\right)^{-0.15} * Pr_p{}^{1/3} \qquad (9)$$

In air coolers, the concept of the actual heat transfer coefficient $\alpha_{p\xi}$ is introduced, which additionally considers the condensation of water vapour (the release of latent heat)

$$\alpha_{p\xi} = \alpha_p * RCJ \qquad (10)$$

The Reynolds number of air needed for Equation (9) is defined as

$$Re_p = \frac{w_o * d_z * \rho_p}{\mu_p} \qquad (11)$$

where $w_o$ denoting the air flow velocity in the smallest free cross-section of the exchanger is determined depending on the arrangement of pipes in the exchanger, as shown in Figure 2.

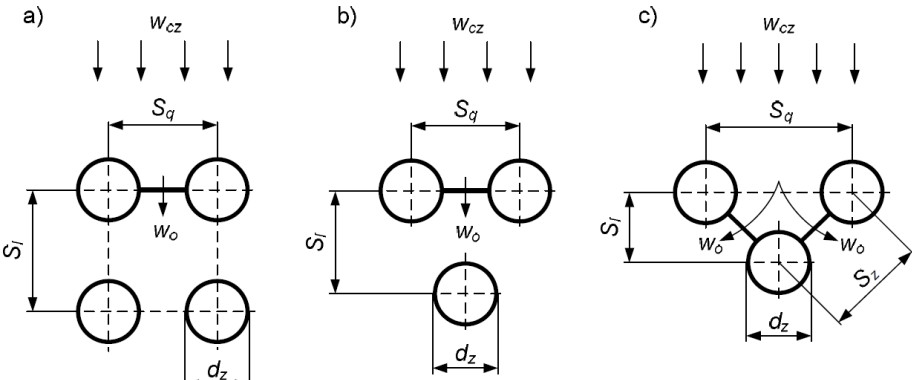

**Figure 2.** Pipe arrangement in the exchanger: (**a**) in-line; (**b**) staggered for $\frac{S_l}{d_z} \geq 0.5 * \left(2 * \frac{S_q}{d_z} + 1\right)^{0.5}$; (**c**) staggered for $\frac{S_l}{d_z} < 0.5 * \left(2 * \frac{S_q}{d_z} + 1\right)^{0.5}$ [16].

The in-line arrangement $w_o$ is defined by Equation (12) and the staggered arrangement is defined by Equation (13).

$$w_o = w_{cz} * \frac{S_q * \xi}{(S_q - d_z) * (\xi - \delta_{\dot{z}})} \tag{12}$$

$$w_o = MAX\left(w_{cz} * \frac{S_q * \xi}{(S_q - d_z) * (\xi - \delta_{\dot{z}})} ; w_{cz} * \frac{0.5 * S_q * \xi}{(S_z - d_z) * (\xi - \delta_{\dot{z}})}\right) \tag{13}$$

The parameter $S_z$ can be calculated as

$$S_z = \left(0.25 * S_q{}^2 + S_l{}^2\right)^{0.5} \tag{14}$$

The coefficient RCJ in Equation (10) (as the ratio of the total heat to the sensible heat of the process) can be calculated from

$$RCJ = 1 + 2480 * \frac{(X_{p1} - X''{}_{pz})}{(t_{p1} - t_z)} \tag{15}$$

Moisture content $X''{}_{pz}$ in saturated air with temperature $t_z$ can be determined from the equations:

$$X''{}_{pz} = 0.622 * \frac{p''{}_w}{p_a - p''{}_w} \tag{16}$$

$$p''{}_w = 610.7 * 10^B \tag{17}$$

$$B = \frac{t_z}{31.6639 + 0.131305 * t_z + 2.63247 * 10^{-5} * t_z{}^2} \tag{18}$$

The average surface temperature can be calculated as the weighted average over the area of fins and pipes between fins

$$t_z = \frac{A_{\dot{z}} * t_{\dot{z}m} + A_r * t_{\dot{z}p}}{A_c} \tag{19}$$

$$t_{\dot{z}m} = t_{pm} - \varepsilon_{\dot{z}} * \left(t_{pm} - t_{\dot{z}p}\right) \tag{20}$$

$$t_{\dot{z}p} = t_{cm} + \frac{\dot{Q}}{A_w} * \left(\frac{1}{\alpha_c} + \frac{\delta_r}{\lambda_r} * \frac{d_w}{d_m} + R_z\right) \tag{21}$$

The next step is to determine the efficiency of the fins $\varepsilon_{\dot{z}}$ in Equation (22). To determine this parameter, one should assume elementary fins (Figure 3) and the weighted

fin height parameter $h_{\dot{z}}$ in Equation (23), the shape of which depends on the designed cross-sections between parallel finned tubes in a bundle. Quantities $m$ and $\varsigma$ used in Equations (22) and (23) are described in Equations (24) and (25).

$$\varepsilon_{\dot{z}} = \frac{\tan h(m * h_{\dot{z}})}{m * h_{\dot{z}}} \tag{22}$$

$$h_{\dot{z}} = \frac{d_z}{2} * (\varsigma - 1) * (1 + 0.35 * ln\varsigma) \tag{23}$$

$$m = \left( \frac{2 * \alpha_{p\xi}}{\delta_{\dot{z}} * \lambda_{\dot{z}}} \right)^{1/2} \tag{24}$$

$$\varsigma = Z_1 * \frac{B}{d_z} * \left( \frac{A}{B} - Z_2 \right)^{1/2} \tag{25}$$

In the above Equation (25), there are two constants $Z_1$ and $Z_2$ and quantities $A$ and $B$ depending on the shape of the elementary fins. The constants take the following values: for rectangular elementary fins $Z_1 = 1.28$; $Z_2 = 0.2$; for hexagonal fins $Z_1 = 1.27$; $Z_2 = 0.3$. The value $A = S_z$, but the value of $B$ depends on spacing: $B = S_q$ if $S_l > 0.5 * S_q$ or $B = 2 * S_l$ if $S_l < 0.5 * S_q$.

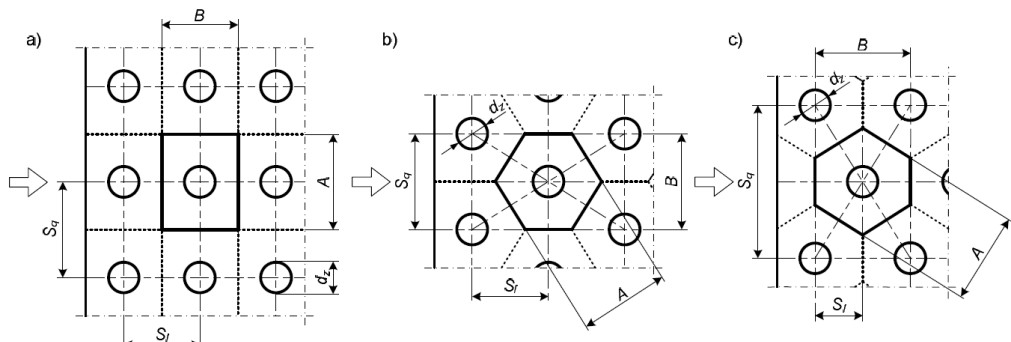

**Figure 3.** Schematic picture of different elementary fins: (**a**) rectangular where $S_q > S_l$; (**b**) hexagonal where $S_l \geq 0.5 * S_q$; (**c**) hexagonal where $S_l < 0.5 * S_q$ [16].

*2.2. Heat Transfer Coefficient from Refrigerant Side $\alpha_c$ and Cooling Capacity of Exchanger $\dot{Q}$*

The next part of the algorithm is the calculation of the heat transfer coefficient from the refrigerant side and cooling capacity of the exchanger. For the analysed problem, there may be two cases (evaporation and condensation) upon which two different calculation algorithms depend.

2.2.1. Condensation

When the vapour of the refrigerant condenses, the Chato formula (26) can be used, which is applicable for horizontal pipes with a known internal diameter [36]

$$\alpha_c = 0.555 * \left( \frac{\lambda'^3_c * \rho'_c * (\rho'_c - \rho''_c) * g * r}{\mu' * (t_k - t_s) * d_w} \right)^{1/3} \tag{26}$$

Additionally, using Newton's dependence (27), Equation (26) can be converted to the form presented in Equation (28) with the applicability range of Re < 35,000.

$$\dot{q}_c = \alpha * (t_k - t_s) \tag{27}$$

$$\alpha_c = 0.456 * \left( \frac{\lambda'^3_c * \rho'_c * (\rho'_c - \rho''_c) * g * r}{\mu' * \dot{q}_c * d_w} \right)^{1/3} \tag{28}$$

With the assumed initial data, the above formula is an implicit function. Zalewski and Niezgoda-Żelasko proposed to apply the dependencies shown in Equations (29)–(34) [16]

$$\dot{Q} = C_4 * \left(1 - \exp\left(-\frac{C_3}{C_1 + C_2 * \dot{Q}^{\frac{1}{3}}}\right)\right) \tag{29}$$

$$C_1 = \frac{\delta_r * A}{\lambda_r * A_m} + R_z + \frac{A}{\alpha_p * (A_r + \varepsilon_{\dot{z}} * A_{\dot{z}})} \tag{30}$$

$$C_2 = \frac{A}{A_w^{4/3} * C} \tag{31}$$

$$C_3 = \frac{A}{\dot{W}_p} \tag{32}$$

$$C_4 = \dot{W}_p * (t_k - t_{p1}) \tag{33}$$

$$C = 0.456 * \left(\frac{\lambda_c'^3 * \rho_c' * (\rho_c' - \rho_c'') * g * r}{\mu' * d_w}\right)^{1/3} \tag{34}$$

In solving Equation (29), we obtain the heat capacity of the condenser for the assumed parameters.

The next step is to determine the air parameters at the exit of the exchanger. The temperature can be calculated by solving Equation (35). Calculations of relative humidity can be calculated from Equations (36)–(39).

$$t_{p2} = t_{p1} + \frac{\dot{Q}}{m_p * c_{pp}} \tag{35}$$

$$\varphi_{p2} = \frac{X_{p1} * p_a}{(0.622 + X_{p1}) * p''_2} \tag{36}$$

$$X_{p1} = 0.622 * \frac{\varphi_{p1} * p''_1}{p_a - \varphi_{p1} * p''_1} \tag{37}$$

$$p''_1 = 610.7 * 10^B \tag{38}$$

$$B = \frac{t_p}{31.6639 + 0.131305 * t_p + 2.63247 * 10^{-5} * t_p^2} \tag{39}$$

### 2.2.2. Evaporation

In the case of the boiling of a refrigerant, the heat transfer coefficient from the boiling thermodynamic medium and the cooling capacity of the evaporator can be determined using the method proposed by Mikielewicz [37]. This correlation is correct for the entire boiling range, i.e., when the liquid–vapour mixture region is changed from 0 to 1 and is based on the Muller-Steinhagen and Heck relationship and uses the Cooper and Dittus–Boelter correlation. Using the above, we obtain the equation

$$\alpha_0 = \alpha_{c0} * \sqrt{R_{MS}^{0.76} + \frac{\dot{q}^{1.34}}{1 + C_p * \dot{q}^{0.6}} * \left(\frac{C_{wo}}{\alpha_{c0}}\right)} \tag{40}$$

The coefficient $\alpha_{c0}$ can be calculated from the Dittus–Boelter correlation

$$\alpha_{c0} = 0.023 * \frac{\lambda'_c}{d'_h} * Re\prime_c^{0.8} * Pr\prime_c^{0.4} \tag{41}$$

where

$$Re'_c = \frac{M_0 * (1 - x_m) * d_w}{\mu'_c} \tag{42}$$

$x_m$ is the average steam dryness fraction, and $M_0$ is the mass flux density of the refrigerant for its complete evaporation and can be calculated by Equation (43).

$$M_0 = \frac{\dot{Q}_0}{r * (1 - x_{c1}) * A_p * n_z} \tag{43}$$

where $\dot{Q}_0$ is the initially assumed heat efficiency of the exchanger, and $A_p$ is the cross-sectional area of the pipe (44), and the average steam dryness fraction, assuming complete conversion to a single-phase fluid, is defined as Equation (45).

$$A_p = \pi * \frac{d_w{}^2}{4} \tag{44}$$

$$x_m = \frac{x_{c1} + 1}{2} \tag{45}$$

By transforming Equations (1)–(3), we obtain Expression (46); with that, it is possible to calculate the product of the heat transfer coefficient and the heat transfer surface area

$$k_{Aw} * A_w = -\dot{W}_p * \ln\left(1 - \frac{\dot{Q}_0}{\dot{W}_p * (t_{p1} - t_o)}\right) \tag{46}$$

The heat transfer surface area can be defined as a function of the heat flux density (47). Substituting Equations (47), (40) and (6) into Expression (46) we obtain Equation (47) with coefficients (49)–(51).

$$A_w = \frac{\dot{Q}_0}{\dot{q}} \tag{47}$$

$$C_4 = \frac{\dot{Q}_0 * \dot{q}^{-1}}{C_2 + C_3 * \left(R_{MS}{}^{0.76} + C_1 * \frac{\dot{q}^{1.34}}{1 + C_p * \dot{q}^{0.6}}\right)^{-0.5}} \tag{48}$$

$$C_1 = \left(\frac{C_{wo}}{\alpha_{c0}}\right)^2 \tag{49}$$

$$C_2 = \frac{\delta_r}{\lambda_r} * \frac{A_w}{A_m} + R_z + \frac{1}{\alpha_{p\zeta 0}} * \frac{A_w}{\varepsilon_{\dot{z}0} * A_{\dot{z}} + A_r} \tag{50}$$

$$C_3 = \frac{1}{\alpha_{c0}} \tag{51}$$

Other parameters necessary to calculate Equation (48) are: the modified Cooper correlation $C_{wo}$ written as Equation (52) and $R_{MS}$, which is a modified Muller-Steinhagen and Heck relationship (54).

$$C_{wo} = 55 * p_{r0}{}^{0.12} * \left(-\log_{10}(p_{r0})\right)^{-0.55} * M_m{}^{-0.5} \tag{52}$$

$$p_{r0} = \frac{p_n}{p_{kr}} \tag{53}$$

$$R_{MS} = \left(1 + 2 * x_m * \left(\frac{1}{f_1} - 1\right)\right) * (1 - x_m)^{\frac{1}{3}} + x_m{}^3 * \frac{1}{f_2} \tag{54}$$

where

$$f_1 = \frac{\rho''_c}{\rho'_c} * \left(\frac{\mu'_c}{\mu''_c}\right)^{0.25} \tag{55}$$

$$f_2 = \frac{\mu''_c}{\mu'_c} * \frac{c'_{pc}}{c''_{pc}} * \left( \frac{\lambda'_c}{\lambda''_c} \right)^{1.5} \tag{56}$$

Next, the correction for the correlation of D. and J. Mikielewicz should be calculated as

$$C_p = 0.00253 * Re'_o{}^{1.17} * \frac{(R_{MS} - 1)}{(r * M_0)^{0.6}} \tag{57}$$

where

$$Re'_o = \frac{M_0 * d_w}{\mu'_c} \tag{58}$$

Solving Equation (59) with respect to $\dot{q}_w$, the desired exchanger capacity is determined from Equation (47), which can be written as

$$\dot{Q}_r = \dot{q}_w * A_w \tag{59}$$

At the same time, with not knowing $\dot{q}_w$, Equation (48) should be modified to obtain the form of a function from which $\dot{q}_w$ can be determined as Equations (60)–(62).

$$(\dot{q}_w) = C_4 - \frac{\dot{Q}_0 * \dot{q}_w{}^{-1}}{C_2 + C_3 * \left( R_{MS}{}^{0.76} + C_1 * \frac{\dot{q}_w{}^{1.34}}{1 + C_p * \dot{q}_w{}^{0.6}} \right)^{-0.5}} \tag{60}$$

$$C_4 = -\dot{W}_p * \ln \left( 1 - \frac{\dot{Q}_0}{\dot{W}_p * (t_{p1} - t_o)} \right) \tag{61}$$

$$\dot{W}_p = m_p * c_{pp} * RCJ \tag{62}$$

When calculating the parameters from the algorithm presented above, attention should be paid to making preliminary assumptions, which must then be confronted with the final calculation values. Depending on the discrepancy of the results, another iteration is conducted until satisfactory calculation results are obtained.

It is recommended to use the temperature of the external heat exchange surface as $t_z = t_o$ for preliminary calculations. After determining the heat flux density, the assumed temperature should be corrected in accordance with Equations (68)–(72). The cooling capacity of exchanger $\dot{Q}$ should also be estimated or taken from the manufacturer's data sheet, if available. Additionally, the elementary equations that will be used to calculate the necessary parameters are presented below (Equations (63)–(72)).

$$d_m = \frac{d_z - d_w}{\ln \left( \frac{d_z}{d_w} \right)} \tag{63}$$

$$\delta_r = \frac{d_z - d_w}{2} \tag{64}$$

$$A_{\dot{z}} = 2 * \left( S_q * S_l - \frac{\pi * d_z{}^2}{4} \right) * \frac{1}{\xi} \tag{65}$$

$$A_r = \pi * d_z * (\xi - \delta_{\dot{z}}) * \frac{1}{t} \tag{66}$$

$$A_c = A_{\dot{z}} + A_r \tag{67}$$

In subsequent iterations, based on the obtained results, the following remaining values are assumed in Equations (40)–(62): final air temperature $t_{p2}$ by Equation (68), average air temperature $t_{pm}$ by Equation (69), average fin temperature: $t_{\dot{z}m}$ by Equation (71) and the temperature of the outer surface of the pipes, $t_{\dot{z}p}$, by Equation (70). Moreover, due to the

assumption of the occurrence of only a phase transformation, the average temperature of the coolant equals the boiling point $t_{cm} = t_o$.

$$t_{p2} = t_{p1} + \frac{\dot{Q}}{m_p * c_{pp} * RCJ} \tag{68}$$

$$t_{pm} = \frac{t_{p1} + t_{p2}}{2} \tag{69}$$

$$t_{\dot{z}p} = t_{cm} + \frac{\dot{Q}}{A_w} * \left( \frac{1}{\alpha_0} + \frac{\delta_r}{\lambda_r} * \frac{d_w}{d_m} + R_z \right) \tag{70}$$

$$t_{\dot{z}m} = t_{pm} - \varepsilon_{\dot{z}} * \left( t_{pm} - t_{\dot{z}p} \right) \tag{71}$$

$$t_{z1} = \frac{A_{\dot{z}} * t_{\dot{z}m} + A_r * t_{\dot{z}p}}{A_c} \tag{72}$$

When the error of the iteration is satisfactorily small, the final parameters of the air leaving the exchanger should be determined. To do this, the following calculations for data obtained from the last iteration are performed, called $\dot{Q}_2$, to determine the moisture content in the air after the exchanger

$$\dot{Q}_{lat} = \dot{Q}_2 - \frac{\dot{Q}_2}{RCJ} \tag{73}$$

By knowing the value of the latent heat taken over by the evaporator from the air, it is possible to calculate the relative humidity at the outlet of the exchanger (Equation (74)), with the value $X_{p2}$ calculated by Equation (75), using Equations (38) and (39).

$$\varphi_{p2} = \frac{X_{p2} * p_a}{(0.622 + X_{p2}) * p_2''} \tag{74}$$

$$X_{p2} = \left( X_{p1} - \frac{\dot{Q}_{lat}}{r_w * m_p} \right) \tag{75}$$

By performing calculations of Equations (68) and (75), we obtain the values of the searched parameters.

## 3. Validation of the Proposed Algorithm

To validate the presented mathematical algorithm, experimental tests were performed on a real air-drying device. The device was tested in a test chamber maintaining constant air parameters at a temperature of 25 °C (±1 °C) and a relative humidity of 60% (±5%). Figure 4 shows the position of the evaporator, the condenser and the circulating fan of the drying device as well as the location of the measuring sensors (six points).

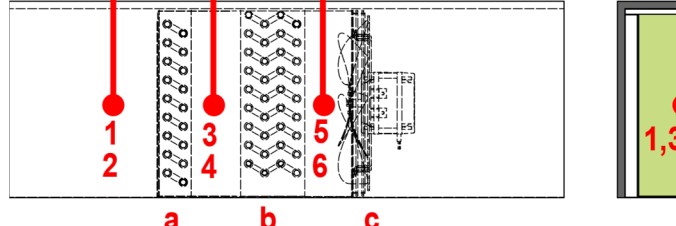 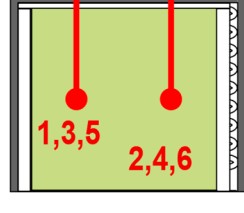

**Figure 4.** Location of measuring points (1–6) and components (from the left): evaporator (**a**), condenser (**b**) and fan (**c**).

The condenser and evaporator are placed in a rectangular duct, which is additionally insulated from the top and sides with a 20-mm layer of extruded polystyrene (XPS) to

minimise the influence of external factors on the readings of air parameters. The parameters in points 1 and 2 are the initial (environment) parameters.

The next figure (Figure 5) shows the heat exchangers with marked flow characteristics through them. The inlets are marked in blue, and the outputs are marked in red.

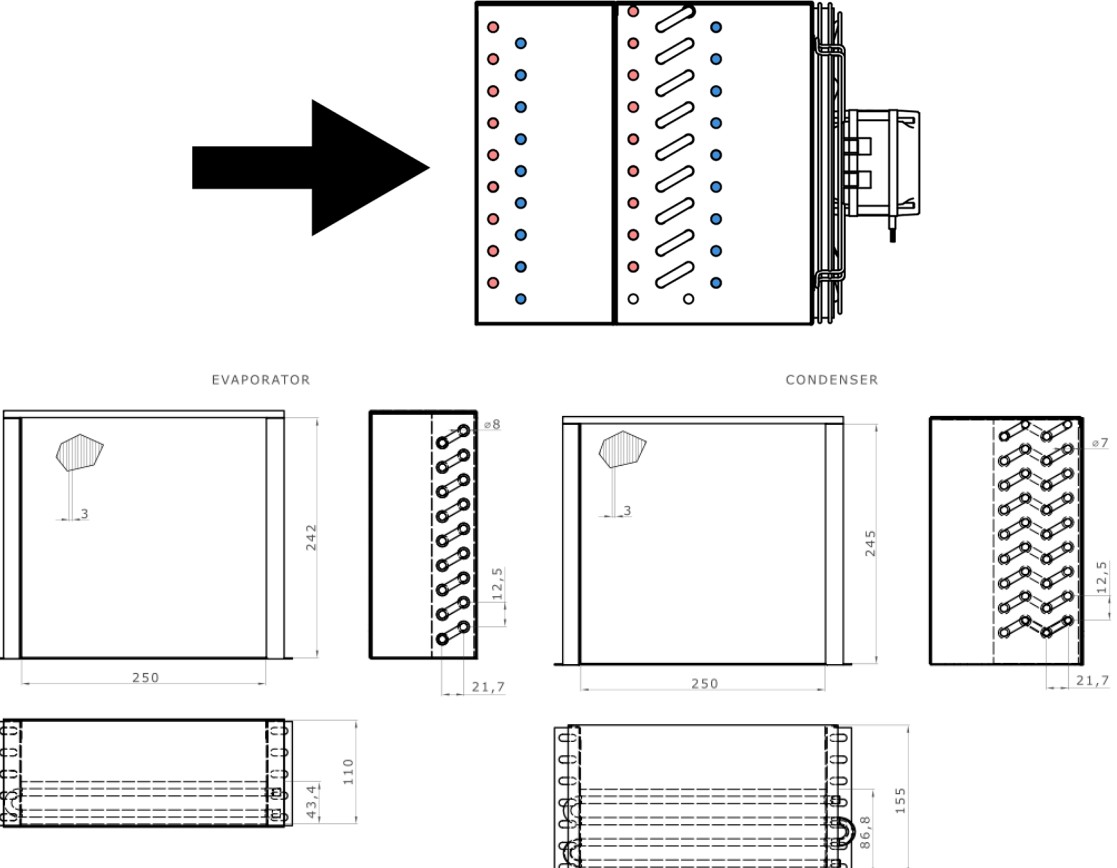

**Figure 5.** Flow characteristics through (from left) the evaporator and the condenser and their geometry.

The exchangers are made of copper pipes $(\lambda_r = 384, \mathrm{W/(mK)})$ with aluminium fins $(\lambda_{\dot{z}} = 220, \mathrm{W/(mK)})$. The pipe system is a staggered arrangement and the flow is countercurrent. Technical data concerning the exchangers are presented in Table 2. The parameter $\dot{Q}_0$ was read from the manufacturer data.

**Table 2.** Data of exchangers [21].

| Evaporator | | | | | | | | | | | |
|---|---|---|---|---|---|---|---|---|---|---|---|
| $G$ | $H$ | $d_z$ | $d_w$ | $S_q$ | $S_l$ | $\xi$ | $\delta_{\dot{z}}$ | $n_r$ | $n_z$ | $N_{rz}$ | $\dot{Q}_0$ |
| 0.25 | 0.2415 | 0.008 | 0.0072 | 0.0125 | 0.0217 | 0.003 | 0.0002 | 9 | 9 | 2 | 1270 |
| Condenser | | | | | | | | | | | |
| $G$ | $H$ | $d_z$ | $d_w$ | $S_q$ | $S_l$ | $\xi$ | $\delta_{\dot{z}}$ | $n_r$ | $n_z$ | $N_{rz}$ | $\dot{Q}_0$ |
| 0.25 | 0.245 | 0.007 | 0.0062 | 0.0125 | 0.0217 | 0.003 | 0.0002 | 9 | 9 | 4 | 3300 |

The calculations ignored the thermal resistance of pollutants on the exchangers $R_z = 0$; furthermore, the following values were adopted for:

Gravity $g = 9.81, \mathrm{m/s^2}$.

Atmospheric pressure $p_a = 101{,}575, \mathrm{Pa}$.

Evaporation and condensation temperatures measured during the tests $t_k = 41.6$, °C and $t_o = 2.5$, °C.

The heat pump operates on the propane refrigerant, R290, the properties of which are presented in Table 3 for the given temperatures. In addition, it contains the properties of the air that are used for the calculations.

**Table 3.** Properties of the refrigerant and humid air [38].

| | Refrigerant | | | Air | |
|---|---|---|---|---|---|
| Parameter | For $t_k$ | For $t_o$ | Parameter | For 25.1, °C ; 57.2, % | For 19.4, [°C ], 72.1, % |
| $\rho'_c$ | 467.00 | 520.43 | $\rho_p$ | 1.18 | 1.22 |
| $\rho''_c$ | 31.56 | 11.18 | $c_{pp}$ | 1009.11 | 1008.86 |
| $\mu'_c$ | $9.90 \times 10^{-5}$ | $1.33 \times 10^{-4}$ | $\lambda_p$ | 0.026 | 0.025 |
| $\mu''_c$ | $9.29 \times 10^{-6}$ | $7.45 \times 10^{-6}$ | $\mu_p$ | $18.42 \times 10^{-6}$ | $17.95 \times 10^{-6}$ |
| $\lambda'_c$ | $8.92 \times 10^{-2}$ | $1.08 \times 10^{-1}$ | $Pr_p$ | 0.72 | 0.72 |
| $\lambda''_c$ | $2.25 \times 10^{-2}$ | $1.70 \times 10^{-2}$ | | | |
| $c'_{pc}$ | 2849.85 | 2466.35 | | | |
| $c''_{pc}$ | 2341.91 | 1813.19 | | | |
| $Pr'_c$ | 3.16 | 3.04 | | | |
| $Pr''_c$ | 0.97 | 0.80 | | | |
| $r$ | $3.04 \times 10^5$ | $3.70 \times 10^5$ | | | |
| $M_m$ | - | 44.10 | | | |
| $\sigma$ | - | $7.02 \times 10^{-3}$ | | | |
| $p_n$ | - | $4.92 \times 10^5$ | | | |
| $p_{kr}$ | - | $4.25 \times 10^6$ | | | |

Chemical and physical data were taken from Refrigeration Utilities program [38]. The air properties are given for the air parameters in the points determined during the tests—before the evaporator (ambient) and between the evaporator and condenser. The measurement parameters are presented in the following sections.

### 3.1. Verification Calculations

Figure 6 schematically shows energy losses or gains in the tested device.

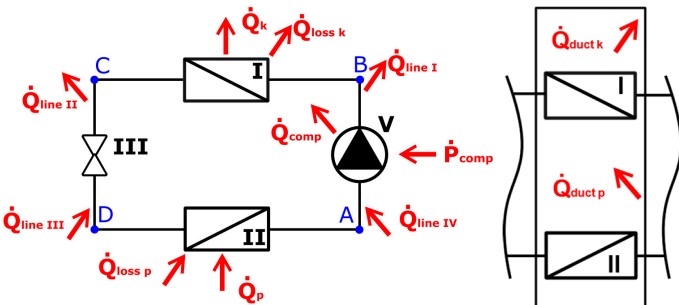

**Figure 6.** Energy losses or gains in the tested device.

Figure 6 shows the measurement points: I—condenser, II—evaporator, III—expansion device, V—compressor. Letter measurement points denote: A—superheat, B—discharge temperature, C—subcooling, D—temperature at the evaporator inlet. Apart from the basic parameters, which are the compressor power $\dot{P}_{comp}$, the cooling capacity of the evaporator

$\dot{Q}_p$ and the condenser $\dot{Q}_k$, there are also unknowns related to the following heat losses: from the compressor crankcase $\dot{Q}_{comp}$, on the pipeline with different temperature $\dot{Q}_{line}$ and the resulting heat exchange by the structural elements of the exchangers $\dot{Q}_{loss}$. Additionally, the losses resulting from heat exchange with the environment should be considered. Although the unit is in an insulated duct, the air exchanges heat after both the $\dot{Q}_{dust\ p}$ evaporator and the $\dot{Q}_{dust\ k}$ condenser, which gives Equation (76). In this equation is also $\dot{Q}_{line}$, which is the heat transfer through the insulated pipeline I, II, III and IV with the environment.

$$\left( \dot{Q}_p + \dot{Q}_{loss\ p} \right) + \dot{P}_{comp} + \left( \dot{Q}_{line\ III} + \dot{Q}_{line\ IV} \right) + \dot{Q}_{duct\ k} = \left( \dot{Q}_k + \dot{Q}_{loss\ k} \right) + \dot{Q}_{comp} + \left( \dot{Q}_{line\ I} + \dot{Q}_{line\ II} \right) + \dot{Q}_{duct\ p} \quad (76)$$

Heat loss from well-insulated connecting tubes was considered to be negligible. The channel insulated with extruded polystyrene also prevented heat transfer from the channel to the environment. The highest energy loss, estimated at 44 W in this case, is in the uninsulated compressor housing. The non-insulated contact points between exchanger casings also influence each other. Since their contact point is irregular and there is a space filled with air between them, the exact heat transfer value is difficult to estimate.

The input data are listed in Tables 2 and 3. The parameters of the air at the inlet to the dryer are assumed to be the parameters in the test chamber. All the calculations in the previous part of the work were made, focusing on the determination of the heat transfer coefficient from the air side, refrigerant, cooling capacity and air outlet parameters. The value of the K parameter used in Equation (9) was adopted for the calculations: 0.33 for the evaporator and 0.38 for the condenser. The constants $Z_1$ and $Z_2$ in Equation (25) assume the same values for each exchanger: for hexagonal ribs $Z_1 = 1.27$ and $Z_2 = 0.3$.

Table 4 (below) shows the results of the first iteration calculations for the assumed exchanger efficiency $\dot{Q}_o = 1270$ W and the temperature of the external surface $t_z = t_o = 2.5\ °C$. The initial data were: $t_{p1} = 25.1\ °C$, $\varphi_{p1} = 57.2\%$ and $t_o = 2.5\ °C$. Additionally, the initial steam dryness fraction was determined for the initial parameters of the factor $x_{c1} = 0.29$.

**Table 4.** Calculations for the first iteration of the evaporator.

| $w_o$ | $Re_p$ | $Nu_p$ | $RCJ$ | $\alpha_{p\xi}$ | $\varepsilon_{\dot{z}}$ | $M_0$ | $Re\prime_c$ | $\alpha_{c0}$ | $C_{wo}$ | $R_{MS}$ | $Re\prime_o$ |
|---|---|---|---|---|---|---|---|---|---|---|---|
| 9.23 | 4730.10 | 35.48 | 1.75 | 202.13 | 0.86 | 13.2 | 253.45 | 45.05 | 6.629 | 20.70 | 713.96 |
| $C_p$ | $C_1$ | $C_2$ | $C_3$ | $C_4$ | $\dot{W}_p$ | $f(\dot{q}_w)$ | $\alpha_0$ | $k_{Aw}$ | $\dot{Q}_r$ | $t_{p2}$ | $\varphi_{p2}$ |
| 0.01 | 0.02 | 0.0008 | 0.02 | 60.58 | 390.72 | 17,151.7 | 2113.63 | 818.2 | 1745.83 | 20.63 | 66.2 |

Due to the initial assumptions, further iterations of the evaporator calculations should be conducted. It was assumed that the criterion that ends the calculations can be expressed as the ratio: $\frac{\left| \dot{Q}_o - \dot{Q}_2 \right|}{\dot{Q}_2} \leq 0.5$ [%]. The critical input and output data together with the result of the criterion calculation are presented in Table 5 (below).

**Table 5.** Iterative calculations of the condenser.

| Iterations | Input Data | | Output Data | | | | Criterion |
|---|---|---|---|---|---|---|---|
| | $\dot{Q}_o$ | $t_z$ | $\dot{Q}_2$ | $t_{p2}$ | $\varphi_{p2}$ | $t_z$ | |
| 1 | 1270 | 2.5 | 1745 | 20.6 | 66.2 | 12.1 | 27.2 |
| 2 | 1745 | 12.1 | 1431 | 20.8 | 68.7 | 11.6 | 22.0 |
| 3 | 1431 | 11.6 | 1532 | 20.6 | 68.9 | 11.6 | 6.6 |
| 4 | 1532 | 11.6 | 1508 | 20.7 | 68.7 | 11.7 | 1.6 |
| **5** | **1508** | **11.7** | **1512** | **20.7** | **68.7** | **11.6** | **0.3** |

The results of the fifth iteration complete the evaporator calculation. The parameters of the air leaving the exchanger are automatically the input parameters of the condenser air. The results of the computational model for the condenser are shown in Table 6.

**Table 6.** Calculations for the condenser.

| $w_o$ | $Re_p$ | $Nu_p$ | $\alpha_p$ | $\varepsilon_{\dot{z}}$ | $C$ | $C_1$ | $C_2$ |
|--------|---------|--------|------------|----------|-----------|--------|----------------------|
| 7.20 | 3424.79 | 34.56 | 123.42 | 0.88 | 40,492.20 | 0.001 | $4.41 \times 10^{-5}$ |

| $C_3$ | $C_4$ | $\dot{W}_p$ | $f\left(\dot{Q}_r\right)$ | $\alpha_k$ | $k_{Aw}$ | $t_{p2}$ | $\varphi_{p2}$ |
|--------|---------|---------|---------|---------|--------|--------|--------|
| 0.0008 | 4666.03 | 222.86 | 1862.23 | 1841.99 | 647.51 | 29.02 | 41.73 |

### 3.2. Test Results

During the tests, Pt 100 temperature probes (range $-100\,^\circ$C to 450 $^\circ$C and accuracy ($\pm0.15 + 0.002 *$ measurement) and relative air humidity sensors EE210 (range 0% to 100 % and accuracy 1.3%) were used. The pressures were measured with Testo 549i BT probes with a measuring range of $-1$ to 60 bar and an accuracy of $\pm0.5\%$. The air flow was measured with a MeasureMe MT891 vane anemometer with a measuring range of 0.4 to 30 m/s, with a resolution of 0.1 and an accuracy of $\pm3\%$. A rectangular duct with a height and width corresponding to the dimensions of the exchanger and a depth of 0.5 m was made in front of the evaporator. In the middle of its depth, three measurements (during each cycle) were made at nine measurement points: three horizontal and three vertical lines. The obtained results were averaged at 3.1 m/s. The test results were averaged and are presented in Figure 7 and Table 7.

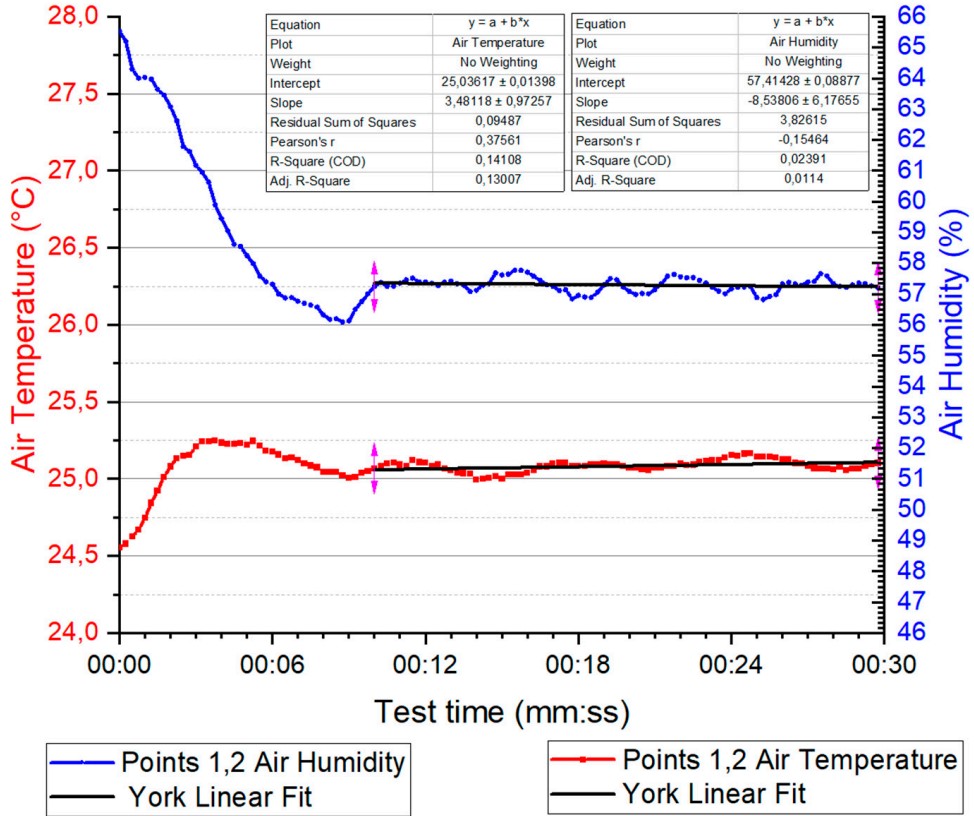

**Figure 7.** *Cont.*

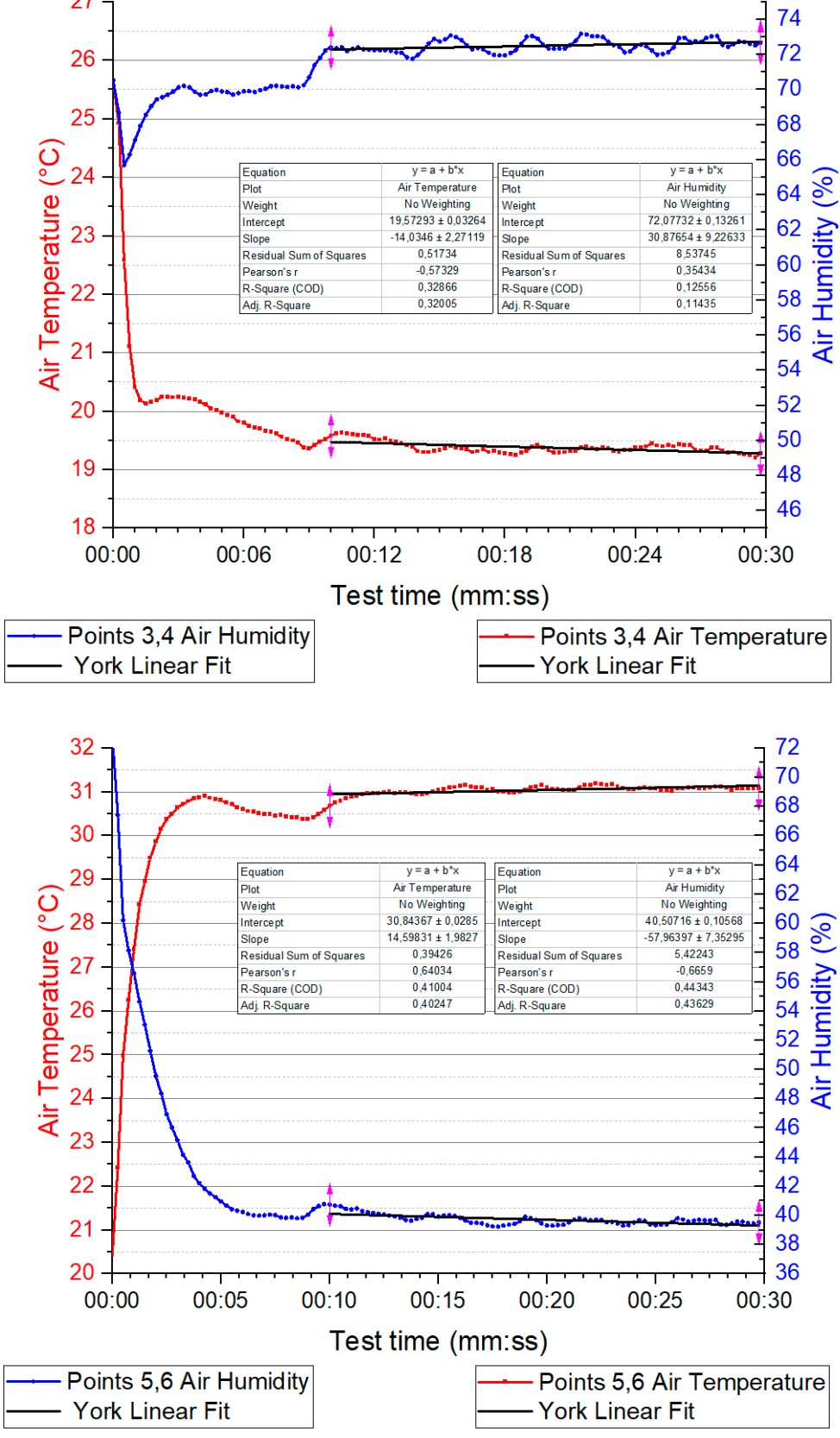

**Figure 7.** Distribution of air temperature and relative humidity at measuring points.

**Table 7.** Averaged measurement values.

| Parameter | Symbol | Value | Enthalpy for the Point |
|---|---|---|---|
| Evaporation pressure | $p_c$ | 13.2 | - |
| Condensation pressure | $p_e$ | 4.1 | - |
| Air flow velocity | $w_p$ | 3.1 | - |
| Air temperature at points 1 and 2 | $t_{p1}$ | 25.1 | 55.2 |
| Air temperature at points 3 and 4 | $\varphi_{p1}$ | 59.2 | |
| Air temperature at points 5 and 6 | $t_{p2}$ | 19.4 | 45.1 |
| Air relative humidity at points 1 and 2 | $\varphi_{p2}$ | 72.1 | |
| Air relative humidity at points 3 and 4 | $t_{p3}$ | 30.9 | 59.4 |
| Air relative humidity at points 5 and 6 | $\varphi_{p3}$ | 39.8 | |
| Heat pump—point A | $t_{cA}$ | 3.4 | 480.6 |
| Heat pump—point B | $t_{cB}$ | 58.3 | 557.4 |
| Heat pump—point C | $t_{cC}$ | 41.2 | 211.6 |
| Heat pump—point D | $t_{cD}$ | 1.7 | 211.6 |

Figure 7 shows the distribution of air temperature and relative humidity at the following points: before the evaporator (averaged 1 and 2), behind the evaporator and before the condenser (averaged 3 and 4) and after the dryer (averaged 5 and 6). All measurements were made during actual operation of the drying heat pump. The condensate was removed and the dried air was returned to the drying chamber. The results were used for model validation with the same assumed conditions.

Table 7 shows the averaged measurements of the values. To omit the initial heating time of the elements, the parameters obtained after ten minutes from each start of the test were taken into account.

### 3.3. Energy Verification of Results

The gas mass flow for the test conditions was read from the compressor manufacturer's data as 18.85 kg/h (0.0052 kg/s). For the given conditions, the cooling capacity is 1645 W and the power consumption is 499 W [39]. From data in Table 7, the calculated condenser capacity is 1798 W and the evaporator capacity is 1399 W. The results of theoretical calculations presented in Tables 5 and 6 differ from the values obtained during the experimental test and are as follows: evaporator capacity = 1512 W and condenser capacity = 1862 W. Moreover, theoretical heat losses to the environment were calculated, which amounted to at least 44 W. Table 8 shows a comparison of the results obtained during the experimental and simulation results. These results are within the accuracy limits of the measurements.

**Table 8.** Comparison of theoretical and empirical results.

| Parameter | Theoretical | Empirical | Relative Error, % |
|---|---|---|---|
| Air temperature at points 3 and 4 | 20.7 | 19.4 | 6.28 |
| Air temperature at points 5 and 6 | 68.7 | 72.1 | 4.95 |
| Air relative humidity at points 3 and 4 | 29.0 | 30.9 | 6.55 |
| Air relative humidity at points 5 and 6 | 41.7 | 39.8 | 4.56 |
| Condenser capacity | 1862 | 1798 | 3.44 |
| Evaporator capacity | 1512 | 1399 | 7.47 |

## 4. Optimisation of Fan Operation

Based on the previously described mathematical algorithm, it is possible to easily determine the heat capacity of exchangers when one or more variables are changed. To optimise the operation of the fan used in the dryer, it is possible to analyse the results for multiple fan operating points—when changing the air speed. Calculations were made for the data presented in the previous chapters at the air velocity range of 1.1 to 4.9 m/s. In Figure 8 the determined characteristics of the exchangers are shown (c). Since an on–off compressor was used, on the basis of catalogue data, for the parameters of the refrigerant during the tests, the cooling capacity of the compressor was marked by the blue line in the graph "c". Optimisation of the air flow should be carried out maximally to the point of intersection of the blue and black lines at the level of 3.25 m/s. For higher speeds, the system is undersized. In addition, the percentage change in the cooling capacity as a function of the percentage change in the air flow velocity "d" is shown.

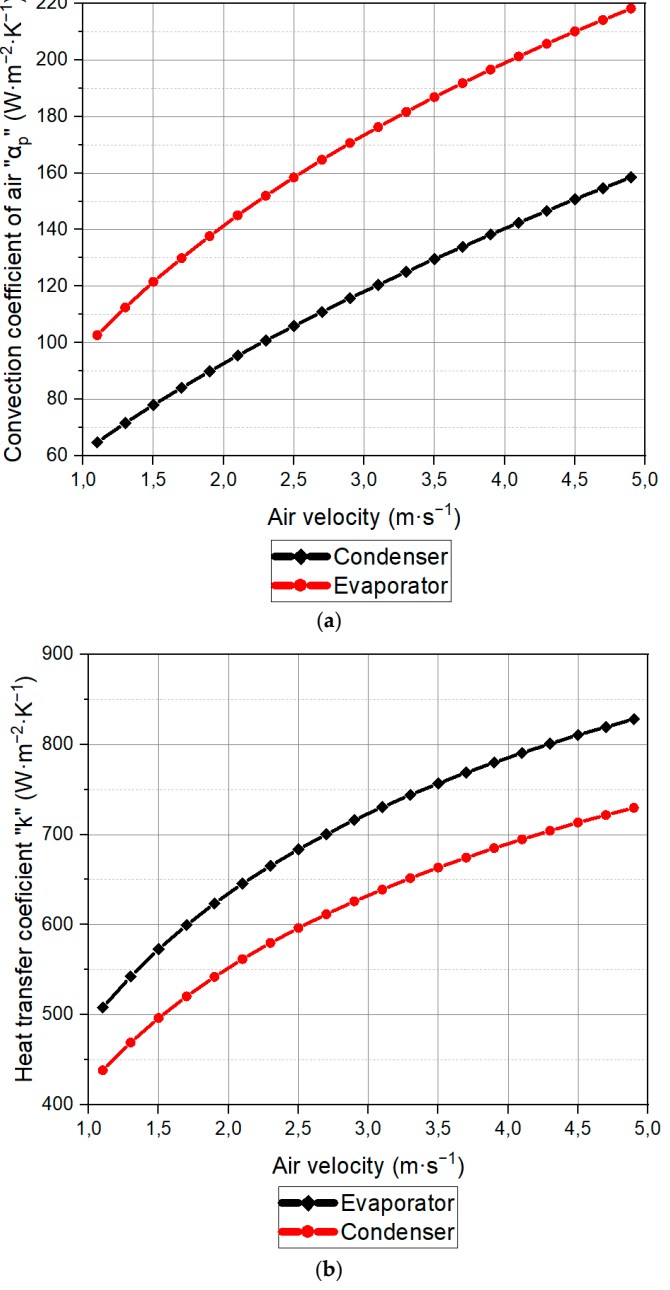

**Figure 8.** *Cont.*

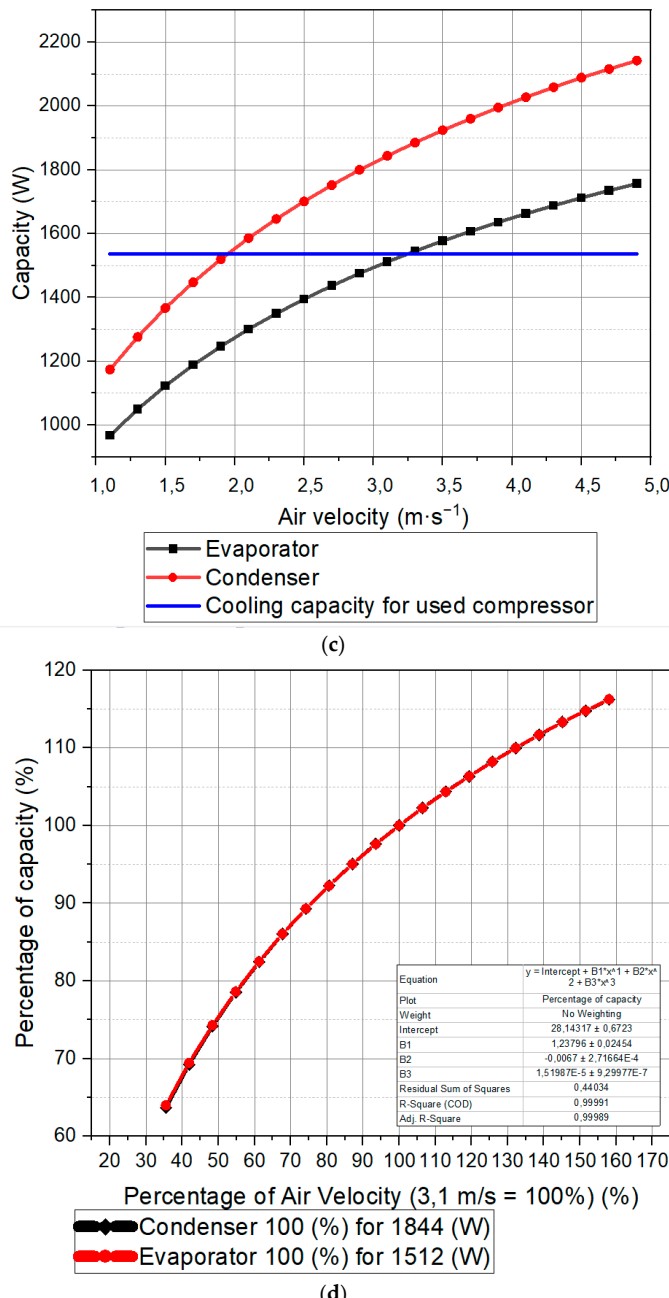

**Figure 8.** Calculation results for different air flow velocities.

The other two charts concern the change in the heat transfer coefficient ($\alpha_p$) "a" and the heat transfer coefficient (exchanger wall) ($k_A$) "b". When air flow velocity increases, the cooling capacity of the heat exchangers also rises. Based on polynomial regression of the third order, the equation of the curve in the "d" chart on Figure 8 is

$$f(x) = 28.1432 + 1.238 * x - 0.0067 * x^2 + 1.520 * 10^{-5} * x^3 \tag{77}$$

while maintaining the coefficient of determination R-square at the level of 0.999991.

The given formula is generalised, although the tests were performed for one type of construction. The research methodology, however, is similar for all types of air-cooled exchangers, so it can be expected that the formula will be similar. The procedure is as follows:

● develop a simulation model of the steady state heat exchanger using well-tested relationships;

- simulate for a variable flow velocity of the air;
- the flow velocity of the air is dependent on the drive and speed of the fan. This characteristic is available from the manufacturer or can be easily measured;
- work out the relationship between exchanger performance and power consumption of the fan drive on the basis of the above simulations;
- this makes it possible to optimize the control of the power consumption.

## 5. Conclusions

This paper presents a mathematical model for the calculation of the condenser and evaporator heat exchangers, which are important components of a heat pump-powered air dryer. The calculation of an actual drying unit with the presented model was conducted, calculating both energy fluxes and heat transfer coefficients. The calculation algorithm is quasistatic, which means that it does not consider the change in air parameters during the drying of the test load. Nevertheless, the results obtained from the validation of the model based on experimental data are consistent and positive. For the end user, it is the total energy consumption of the unit that is more important than the thermodynamically determined COP, which only includes the power of the compressor and not the associated equipment (fan), as the operation of the fan affects the total energy consumed by the unit. The presented model was used to determine the characteristic curve relating the heat exchanger performance to the fan performance. This equation is a necessary building block for optimising the entire unit to reduce energy consumption.

The fan capacity has a dominant influence on the air side convection heat transfer coefficient (100–220%), both in the case with and without condensation. As shown, by means of fan control, the capacity of the exchangers can be varied in a range of 60–120%, adapting to the current heat capacity of the heat pump. When using a fan with a frequency converter, the speed can be adjusted, minimizing the power consumed by the fan according to the current need of the heat exchanger. This requires control based on the developed characteristic shown in Figure 8, described by Equation (77). Such characteristics for any heat exchanger, can be obtained using well-verified heat penetration formulas and heat exchanger design methodologies. Reducing the capacity, i.e., the fan speed, almost directly linearly reduces the energy consumption of the fan electric motor.

**Author Contributions:** Conceptualisation, T.M. and P.C.; methodology, formal analysis, investigation, writing—review and editing, and supervision, P.C.; validation, resources, data curation, writing—original draft preparation and visualisation, T.M. All authors have read and agreed to the published version of the manuscript.

**Funding:** This research received no external funding.

**Institutional Review Board Statement:** Not applicable.

**Informed Consent Statement:** Not applicable.

**Data Availability Statement:** The data presented in this study are available on request from the corresponding authors.

**Acknowledgments:** Special thanks to JUKA company for the technical support and sharing of their climate chamber for experimental tests.

**Conflicts of Interest:** The authors declare no conflict of interest.

## Nomenclature

| | |
|---|---|
| $A$ | surface area, $m^2$ |
| $c_p$ | specific heat capacity at constant pressure, $\frac{J}{kg \cdot K}$ |
| $D = d$ | diameter, m |
| $G$ | exchanger width, m |
| $g$ | gravitational acceleration, $\frac{m}{s^2}$ |
| $H$ | exchanger height, m |

| | |
|---|---|
| $h_{\dot{z}}$ | weighted fin height, m |
| $h$ | enthalpy, $\frac{\text{kJ}}{\text{kg}}$ |
| $k_A$ | heat transfer coefficient related to surface A, $\frac{\text{W}}{\text{m}^2 \cdot \text{K}}$ |
| $L$ | length of heat exchanger pipes, length of flow, m |
| $l$ | length, characteristic dimension, m |
| $\dot{M}$ | mass flux density, $\frac{\text{kg}}{\text{m}^2 \cdot \text{s}}$ |
| $M_M$ | molar mass, $\frac{\text{kg}}{\text{kmol}}$ |
| $\dot{m}$ | mass flow, $\frac{\text{kg}}{\text{s}}$ |
| $N_{RZ}$ | number of rows |
| $NTU$ | number of transfer units |
| $n_r$ | number of pipes |
| $n_z$ | number of injections |
| $n_{\dot{z}}$ | number of fins |
| $p_a$ | atmospheric pressure, Pa |
| $p_n$ | saturation pressure, Pa |
| $p_w$ | partial pressure of water-vapour molecules in the air, Pa |
| $\dot{Q}$ | heat transfer coefficient of the exchanger, W |
| $\dot{q}$ | heat flux, $\frac{\text{W}}{\text{m}^2}$ |
| $R_z$ | thermal resistance of pollutants, $\frac{\text{m}^2 \cdot \text{K}}{\text{W}}$ |
| $r$ | heat of vaporisation, $\frac{\text{J}}{\text{kg}}$ |
| $S_l$ | spacing longitudinal |
| $S_q$ | spacing transversal |
| $S_z$ | spacing diagonally |
| $t$ | temperature, °C |
| $\dot{W}$ | heat capacity of medium flux, $\frac{\text{W}}{\text{K}}$ |
| $w$ | velocity, $\frac{\text{m}}{\text{s}}$ |
| $X_p$ | moisture content in the air, $\frac{\text{kg}_{H_2O}}{\text{kg}_{\text{dry}}}$ |
| $x_c$ | steam dryness fraction |
| Greek Symbols: | |
| $\alpha$ | heat transfer coefficient, $\frac{\text{W}}{\text{m}^2 \cdot \text{K}}$ |
| $\delta$ | thickness, m |
| $\varepsilon_{\dot{z}}$ | fins efficiency |
| $\lambda$ | thermal conductivity, W/(m·K) |
| $\mu$ | dynamic viscosity, kg/(m·s) |
| $\rho$ | density, kg/(m·s) |
| $\sigma$ | surface tension, N/m |
| $\varphi_P$ | relative air humidity, % |
| $\xi$ | fin's pitch, m |
| Subscripts: | |
| 1 | start/inlet value |
| 2 | end/outlet value |
| $c$ | refrigerant |
| $cz$ | frontal area of exchanger |
| $k$ | condensation |
| $lat$ | latent |
| $m$ | average value |
| $o$ | evaporation |
| $p$ | air |
| $r$ | pipe |
| $s$ | wall |
| $w$ | inner surface |
| $z$ | outer surface |
| $\dot{z}$ | fin |
| $'$ | liquid phase in a saturated state |
| $''$ | gas phase in a saturated state |

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
