# Peer review of "Mathematical Model of Air Dryer Heat Pump Exchangers"

_energies, doi:10.3390/en15197092_

Round 1
Reviewer 1 Report
1) Abstract
The abstract was written correctly. It describes all items included in the study. It allows finding out what the authors do and what problem they solved.
2) Introduction
The introduction to the paper presents the problem that appears in air dehumidifiers. The authors reviewed the current literature on the subject matter. They paid particular attention to the state of the experimental research and theoretical calculations of the elements of the air dryer installation, including lamellar heat exchangers. The particular usefulness of the e-NTU method in calculating exchangers has been shown. No comments are made in this regard.
3) Body part of paper
Chapter 3 proposes a calculation algorithm for lamella heat exchangers for air cooling and heating. The assumptions for the model were correctly given. The authors adopted the calculation dependencies for thermal and flow parameters based on the available literature. These dependencies are known and commonly used in this type of calculation. Nevertheless, they are needed to correctly calculate the design calculations of evaporators and condensers used in heat pump air dryers.
In Chapter 4, the developed calculation algorithm was validated. Experimental research was carried out in natural operating conditions. Many geometric, thermal, and flow parameters characterizing the tested exchangers were given. The results of the research are also given in detail. The validation of the model in industrial conditions was successful.
Chapter 5 covers optimizing the operation of the air dryer fan. The total energy consumption of the unit was determined, which includes the power of the compressor and associated equipment, including the fan. The presented model was used to determine the characteristic curve connecting the efficiency of the heat exchanger with the efficiency of the fan. This is very important in optimizing the entire unit to reduce energy consumption. No comments are made in this regard.
4) Conclusion
The conclusions presented are very general and written in a similar way to the executive summary. It is suggested to prepare them in more detail and present the most important achievements of this paper in points, together with rational quantities in numerical form. They will be a guide for future designers and operators of air dryers working on the heat pump principle.
5) Reference
29 items of literature were used, including many articles from recent years. No comments are made in this regard.
Author Response
Reviewer 1
The authors would like to thank the Reviewer for the insightful comments which contributed to the value of the article.
Comments and Suggestions for Authors
1) Abstract
The abstract was written correctly. It describes all items included in the study. It allows finding out what the authors do and what problem they solved.
2) Introduction
The introduction to the paper presents the problem that appears in air dehumidifiers. The authors reviewed the current literature on the subject matter. They paid particular attention to the state of the experimental research and theoretical calculations of the elements of the air dryer installation, including lamellar heat exchangers. The particular usefulness of the e-NTU method in calculating exchangers has been shown. No comments are made in this regard.
3) Body part of paper
Chapter 3 proposes a calculation algorithm for lamella heat exchangers for air cooling and heating. The assumptions for the model were correctly given. The authors adopted the calculation dependencies for thermal and flow parameters based on the available literature. These dependencies are known and commonly used in this type of calculation. Nevertheless, they are needed to correctly calculate the design calculations of evaporators and condensers used in heat pump air dryers.
In Chapter 4, the developed calculation algorithm was validated. Experimental research was carried out in natural operating conditions. Many geometric, thermal, and flow parameters characterizing the tested exchangers were given. The results of the research are also given in detail. The validation of the model in industrial conditions was successful.
Chapter 5 covers optimizing the operation of the air dryer fan. The total energy consumption of the unit was determined, which includes the power of the compressor and associated equipment, including the fan. The presented model was used to determine the characteristic curve connecting the efficiency of the heat exchanger with the efficiency of the fan. This is very important in optimizing the entire unit to reduce energy consumption. No comments are made in this regard.
4) Conclusion
The conclusions presented are very general and written in a similar way to the executive summary. It is suggested to prepare them in more detail and present the most important achievements of this paper in points, together with rational quantities in numerical form. They will be a guide for future designers and operators of air dryers working on the heat pump principle.
According to this suggestion the following passage is added in “Conclusions”
“The fan capacity has a dominant influence on the air-side convection heat transfer coefficient (100-220%), both in the case with and without condensation. As shown, by means of fan control, the capacity of the exchangers can be varied in a range of 60-120%, adapting to the current heat capacity of the heat pump. When using a fan with a frequency converter, the speed can be adjusted, minimising the power consumed by the fan according to the current need of the heat exchanger. This requires control based on the developed characteristic shown in Figure 8, described by equation (77). Such characteristics for any heat exchanger, can be obtained using well-verified heat penetration formulas and heat exchanger design methodologies. Reducing the capacity, i.e. the fan speed, almost directly linearly reduces the energy consumption of the fan electric motor.”
5) Reference
29 items of literature were used, including many articles from recent years. No comments are made in this regard.

Reviewer 2 Report
This article presents a mathematical model of heat pump exchangers and their thermal interaction with a fan for an air dryer. From an industrial dehumidifier test bench this modelling algorithm is verified. In my opinion, this article is of scientific and engineering interesting. And, the quality of this manuscript is good in a total. Therefore, I would like to recommend its acceptation after minor revisions.
1. Although the methodology for modelling is generalized, a general flow chart for modelling is suggested to be briefly described in a figure. In addition, “The research methodology, however, is similar for all types of air-cooled exchangers, so it can be expected that the formula will be similar” is needed to explain extensively.
2. Figures that reflect the geometry of current heat exchangers are suggested to add, including the detailed parameters. Because the specific values involved in the mathematical model are associated with the heat exchanger geometric feature.
3. Section “Heat transfer coefficient from refrigerant side and cooling capacity of exchanger” should be 2.2 but not 2.1. Please check the whole manuscript carefully.
Author Response
Reviewer 2
The authors would like to thank the Reviewer for his detailed review of the article, which allowed for its better elaboration.
Comments and Suggestions for Authors
This article presents a mathematical model of heat pump exchangers and their thermal interaction with a fan for an air dryer. From an industrial dehumidifier test bench this modelling algorithm is verified. In my opinion, this article is of scientific and engineering interesting. And, the quality of this manuscript is good in a total. Therefore, I would like to recommend its acceptation after minor revisions.
- Although the methodology for modelling is generalized, a general flow chart for modelling is suggested to be briefly described in a figure. In addition, “The research methodology, however, is similar for all types of air-cooled exchangers, so it can be expected that the formula will be similar” is needed to explain extensively.
The following passage explaining the methodology is added to the text:
“The procedure is as follows:
- develop a simulation model of the steady-state heat exchanger using well-tested relationships,
- simulate for a variable flow velocity of the air,
- the flow velocity of the air is dependent on the drive and speed of the fan. This characteristic is available from the manufacturer or can be easily measured,
- work out the relationship between exchanger performance and power consumption of the fan drive on the basis of the above simulations.
This makes it possible to optimise the control of the power consumption.”
- Figures that reflect the geometry of current heat exchangers are suggested to add, including the detailed parameters. Because the specific values involved in the mathematical model are associated with the heat exchanger geometric feature.
The geometry of the exchanger has been dimensioned in the revised drawing (Fig. 5).
- Section “Heat transfer coefficient from refrigerant side and cooling capacity of exchanger” should be 2.2 but not 2.1. Please check the whole manuscript carefully.
The entire manuscript was re-checked and revised.

Reviewer 3 Report
The Authors should clarify the novelty of the proposed mathematical model.
Moreover, the model verification should be carried out also when the condensation of the water vapor occurs.
1) Introduction
This section can be improved by including references that are more relevant and by clarifying the novelty of the mathematical model proposed in the present study.
2) Body part of paper
The verification of the proposed model has been carried out by considering a relative humidity of 60% (± 5%). It is more interesting to verify the model for values of the relative humidity higher than 60% when the condensation of the water vapor may occur.
3) Reference
This section can be improved by including references that are more relevant
Author Response
Reviewer 3
The authors would like to thank the reviewer for his elaborate review of the article, which helped to clarify the issues vaguely presented.
The Authors should clarify the novelty of the proposed mathematical model.
The main innovation of the article is the link in the form of a function between exchanger performance and fan energy consumption. For this purpose, well-tested relationships were used to calculate the heat convection coefficients.
Therefore following passage is introduced into the text:
“The innovation of the presented method lies in the fact that, based on well-verified calculation formulas for convective heat transfer on the air side, a relationship has been developed that relates fan power and heat exchanger performance under various operating conditions. The presented method allows optimisation of the entire unit, i.e. adjustment of the ventilator power to the actual heat exchanger capacity. In this way, electricity consumption for the drive is reduced while maintaining the drying capacity of the heat pump.”
Moreover, the model verification should be carried out also when the condensation of the water vapor occurs.
We would like to thank the Reviewer for rightly pointing out that, for drying devices, heat transfer during condensation of the removed water on the external surface of the exchanger tubes is very important. This is exactly the condition under which the model was measured and verified (Figure 7). For the condensing conditions, the air-side convection coefficient relationships were also selected.
For clarification, the following text is added below Figure 7:
“All measurements were made during actual operation of the drying heat pump. The condensate was removed and the dried air was returned to the drying chamber. The results were used for model validation with the same assumed conditions.”
1) Introduction
This section can be improved by including references that are more relevant and by clarifying the novelty of the mathematical model proposed in the present study.
Ten additional items of literature are reviewed in the introduction.
“The effect of circulating air volume on the circulating air temperature, coefficient of performance, moisture extraction rate and specific moisture extraction rate were experimentally investigated in [26]. Among other things, the results showed the effect of air circulation on energy consumption. The energy effects associated with crop drying and the influence of air volume were also pointed out by the authors [27-29]. The authors [30] showed the effect of compressor capacity on drying effects, but did not address the problems of other flow machines. In [31] the drying kinetics and performance of the open, closed and partially open heat pump drying systems were compared. In [32] the novel technology of an air cycle heat pump drying is presented with comparison, simulation and experimental investigation. In [33,34] multistage heat pump systems for drying are presented. The authors [35] analyse drying with and without recirculation which also regulates the airflow to some extent, although without affecting the fan power. In each case, the Authors of the mentioned works pay attention to the energy consumption and the influence of the dried air flow on the process. However, they do not refer to the possibility of controlling the fan and linking it to the operation of the heat exchanger.”
2) Body part of paper
The verification of the proposed model has been carried out by considering a relative humidity of 60% (± 5%). It is more interesting to verify the model for values of the relative humidity higher than 60% when the condensation of the water vapor may occur.
As mentioned earlier, both simulation calculations and experimental studies were carried out with actual dehumidification and therefore condensation of the vapour contained in the air on the outer surface of the evaporator tubes. The relative humidity of the air entering the exchanger exceeded 70%, as can be seen in Figure 7. On the surface of the evaporator, the moisture from the air condensed and the dried air was returned to the drying chamber.
3) Reference
This section can be improved by including references that are more relevant
An additional 10 items of literature on drying heat pumps from recent years have been added to the literature analysis and a brief description of these has been inserted into the text.

Round 2
Reviewer 3 Report
The paper can be published.